# Regulation of Microbial Activity Rates by Organic Matter in the Ross Sea during the Austral Summer 2017

**DOI:** 10.3390/microorganisms8091273

**Published:** 2020-08-21

**Authors:** Renata Zaccone, Cristina Misic, Filippo Azzaro, Maurizio Azzaro, Giovanna Maimone, Olga Mangoni, Gianna Fusco, Alessandro Ciro Rappazzo, Rosabruna La Ferla

**Affiliations:** 1Institute of Polar Sciences, National Research Council (ISP-CNR), 98122 Messina, Italy; filippo.azzaro@cnr.it (F.A.); maurizio.azzaro@cnr.it (M.A.); giovanna.maimone@cnr.it (G.M.); rappale@libero.it (A.C.R.); rosabruna.laferla@cnr.it (R.L.F.); 2Department of Earth, Environment and Life Sciences, University of Genova, 16132 Genova, Italy; cristina.misic@unige.it; 3Department of Biology, University “Federico II” of Naples, 80126 Naples, Italy; olga.mangoni@unina.it; 4Department of Science and Technology, University “Parthenope” of Naples, 80143 Naples, Italy; giannetta.fusco@uniparthenope.it

**Keywords:** Antarctic Ocean, biogeochemical cycles, prokaryotes, enzymatic activities, particulate organic matter

## Abstract

The active prokaryotic communities proliferate in the ecosystems of the Antarctic Ocean, participating in biogeochemical cycles and supporting higher trophic levels. They are regulated by several environmental and ecological forcing, such as the characteristics of the water masses subjected to global warming and particulate organic matter (POM). During summer 2017, two polynyas in the Ross Sea were studied to evaluate key-microbiological parameters (the proteasic, glucosidasic, and phosphatasic activities, the microbial respiratory rates, the prokaryotic abundance and biomass) in relation to quantitative and qualitative characteristics of POM. Results showed significant differences in the epipelagic layer between two macro-areas (Terra Nova Bay and Ross Sea offshore area). Proteins and carbohydrates were metabolized rapidly in the offshore area (as shown by turnover times), due to high enzymatic activities in this zone, indicating fresh and labile organic compounds. The lower quality of POM in Terra Nova Bay, as shown by the higher refractory fraction, led to an increase in the turnover times of proteins and carbohydrates. Salinity was the physical constraint that played a major role in the distribution of POM and microbial activities in both areas.

## 1. Introduction

The Antarctic Ocean encompasses several different ecosystems (winter and summer polynyas, marginal ice zones (MIZ), ice-covered waters, coastal areas, and open sea), subjected to extreme conditions due to low temperatures, seasonal shifts related to high solar radiation and UV exposure in summer, as well as changes in ice cover [1]. Despite these harsh environmental characteristics, several areas such as spring and summer polynyas and MIZs are biological hot-spots that support high rates of primary production [2,3,4], due to a combination of both high light and high nutrient levels, especially iron [5]. These regions significantly contribute to the regulation of the climate system by sequestering the anthropogenic CO_2_ by intense biological productivity and production of Antarctic bottom water (AABW) [2].

In the dynamic and heterogeneous ecosystems of the Antarctic Ocean, diverse and active prokaryotic communities proliferate, participating in biogeochemical cycles and supporting higher trophic levels [3,6]. Several environmental and ecological forces regulate the prokaryotic abundance and activity [7], so that microbial metabolic processes are highly variable amongst stations and depths [8]. Temperature-salinity characteristics of the water masses were often found to control the distribution of microbial communities and several authors found that the degradation processes can be related to specific water masses, suggesting a relationship between microbial metabolism and age/origin of water masses [9,10]. Concerning temperature, the global warming of the polar oceans could favor marine microbes, allowing them to become more active and, consequently, more organic matter (OM) could be consumed [11], despite a rise of temperature would affect the activity of that part of extracellular hydrolytic enzymes (exoenzymes) that are not protected by a tight link to the bacterial cell membrane [12,13].

In addition, both pressure and oxygen concentrations, the latter depending on photosynthetic processes, had a significant effect on the bacterial communities [10] that showed significant differences in abundance, composition, and activity related to the stratification of resources with depth. In most areas, salinity is a significant factor negatively affecting the prokaryotic abundance and enzyme activity. The salinity changes due to ice melt were found to cause rapid shifts in the microbial community, also due to the input of ice-associated microbes into the planktonic community [9,14]. A biogeochemical model implemented in the Ross Sea—as a tool to examine the mechanisms involved in organic matter flux along the water column—suggested that the observed fluxes were due to the timing of production at surface layer and to the size-distribution and quality of the biogenic material of autochthonous origin [15].

Despite previous investigations indicated that heterotrophic prokaryotes use a small fraction (<10%) of primary production, prokaryotes were often strongly related to chlorophyll-*a* concentrations and phytoplankton production [1,16]. Seasonal dynamics of heterotrophic microbial communities generally follow phytoplankton blooms that generate pulses of labile OM. Thus, the heterotrophic prokaryotic activity is regulated by the bioavailability of OM, both dissolved (DOM) [17] and particulate (POM) [1,3].

Prokaryotes assimilate the DOM and convert it into biomass, using the gained materials and energy for their metabolic processes (for instance respiration) [18]. Prokaryotes can also grow on POM, where they found favorable environmental conditions, besides OM availability. Differently than in temperate seas, in the aphotic Ross Sea (RS) about 63% of biogenic organic carbon remineralized by respiration derived from the POC pool, highlighting POC source as the main organic fuel of the biological pump [19].

Prokaryotes can uptake only molecules < 600 Da, therefore extracellular enzyme activity (EEA) for hydrolyzing high molecular weight DOM into low molecular weight compounds is needed [12,20]. EEA controls the rates of decomposition of organic matter and nutrient mineralization, essentially under low-resource conditions [10,13]. EEA is typically induced only under specific circumstances by microbial cells; as an example, EEA has been used for evaluating the bacterial response to ice meltwaters, since enzymes respond rapidly, directly or through changes in microbial community composition, to changes of nutrient and substrate availability [11,13].

The RS is one of the most productive areas of the Southern Ocean [19,21] and includes many marine ecosystems that contribute differently to primary production processes with large interannual variations [4]. These ecosystems make the RS a “laboratory” for the study of the potential effects of global change on the biological cycles and ecological characteristics of the Southern Ocean. In this framework, with the aim to protect ecosystem processes, conserve biodiversity, and promote research on the marine living resources (https://www.ccamlr.org/), the Marine Protected Area of the RS came into force in December 2017 [8].

Most of the research focused on the upper layers of the water column, confirmed the aforementioned close link between the OM availability and the prokaryotic activity response also due to the quick (timescales of days) acclimation of prokaryotes to POM quality changes to sustain their metabolic requirements [12].

In the summer polynya of the RS, prokaryotic activity in terms of electron transport system activity (ETSa) showed a greater dependency on POM availability than on hydrological conditions [22]. In particular, ETSa increases were related to a good trophic value of POM, i.e., the OM showing a dominance of proteins and N over structural carbohydrates and organic C [23]. The N-containing fractions are more attractive to prokaryotic metabolism, as N-rich proteins are used more quickly than carbohydrates because N is the most limiting element for heterotrophic nutrition [23].

In the Artic polar zone, carbohydrates are a more important organic source for prokaryotes [20]. In addition, a close relationship between the EEA pool and POM composition was found [23]. The preferential expression of one enzyme gives evidence on OM quality in a particular environment [9,24].

To gather insight into the prokaryotic community characteristics and activity, and the relationships with POM availability, two summer polynyas of RS were studied in January-February 2017, focusing on the surface 100 m layer. In particular, the metabolic activity of prokaryotes involved in the biogeochemical cycles was investigated b: the total enzymatic hydrolysis rates (focusing on proteasic, glucosidasic, and phosphatasic activities), the microbial respiratory rates (ETS) together with the prokaryotic abundance and biomass. Quantitative and qualitative characteristics of POM were examined.

We aimed at finding out the potential regulation of prokaryotic variables by POM availability and quality, and relate these findings to the environmental constraints in different areas. Through the analysis of turnover times of POM we identified the characteristics of two different macro-areas. We provide evidence of environmental changes on microbial community metabolism.

## 2. Material and Methods

### 2.1. Study Area and Sampling

The spatial variability of microbial parameters between summer polynyas was investigated, focusing on the macro-area of the coastal Terranova Bay polynya (TNB) and the macro-area of the offshore Ross Sea polynya (RS) (Figure 1). From 13 January 2017 to 11 February 2017, a total of 24 stations were sampled for the microbiological variables at different depths. Eighteen stations were in common with the sampling for the particulate organic matter (POM) variables; station 15 was repeated 3 times and station 57 was sampled 2 times.

The two identified macro-areas (coastal TNB and offshore RS) were further divided into sub-areas by a geographical point of view (Table 1 and Figure 1), in accordance with previous investigations [23] that found frontal structures within TNB and RS. The TNB coastal area was divided into three transects: western (TNB-W) stations (13, 14, 15, and the repeated 39 and 81, all placed coastward), TNB-mid stations (18, 19, and 20) placed in the area of the frontal structure previously observed by [23] and eastern (TNB-E) stations (21, 22, and 23), placed in the most open area. The offshore RS macro-area was divided into a mid-sub-area (RS-mid, stations 43, 44, 45), a southern sub-area (RS-S, stations 46, 47, 48, 49, and 50), and a northern sub-area (RS-N, stations 53, 55, 57, 59, and 80) (Table 1). The RS-S and RS-N stations were placed in two areas that, in the previous investigations [23], were separated by a frontal structure.

The sampling was mainly focused on the first 100 m, were four depths were examined: surface, intermediate 1 and 2, 100 m. The two intermediate depths were chosen depending on the water column structure and were slightly different among the two macro-areas (on average, in the TNB coastal macro-area, they were −22 ± 7 m and −61 ± 6 m for intermediate 1 and intermediate 2, respectively; in the offshore RS macro-area, they were −30 ± 8 m and −58 ± 10 m, respectively). Some stations belonging to the TNB area were sampled also in the mesopelagic layer (Table A1).

### 2.2. Environmental and Biochemical Parameters

A rosette sampler was equipped with a Sea Bird Electronics SBE-911plus CTD profiler, to acquire temperature (T), salinity (S), and dissolved oxygen concentration (OX) along the water column. Niskin bottles (volume 20 L) were placed on the same frame, to collect seawater at discrete depths.

For the POM, determinations seawater was filtered through Whatman GFF glass-fiber filters (nominal pore size 0.45 µm) in duplicate. The filters were immediately frozen until analysis in laboratory.

Particulate organic carbon (POC) and particulate nitrogen (PN) were analyzed following Hedges and Stern [25], using a Carlo Erba Mod. 1110 CHN Elemental Analyzer. The samples were acidified with HCl fumes to remove inorganic carbon, and cyclohexanone 2–4-dinitrophenyl hydrazone (purchased from Sigma-Aldrich, Steinheim, Germany) was used as a reference standard.

Particulate protein and carbohydrate concentrations were determined following [26,27], respectively. Albumin and glucose solutions were used to calibrate a Jasco V530 spectrophotometer.

The C content of proteins and carbohydrates were estimated using 0.49 µg C µg protein^−1^ and 0.40 µg C µg carbohydrate^−1^, respectively [28].

### 2.3. Microbiological Measurements

The total Prokaryotic Abundance (PA) was estimated on fixed samples (formaldehyde 2% final), stained with DAPI (4,6-diamidino-2-phenyl-indole) dye [29] and analyzed under a Zeiss AXIOPLAN 2 Imaging microscope; an Axiocam digital camera was used for the Prokaryotic biomass (PB) determination [30]. The contribution of the PB to POC was calculated.

For the potential estimates of total EEA rates, the fluorogenic substrates method was applied [31]. For determination of Alkaline Phosphatase (AP) and β-Glucosidase (GLU) activities, the substrate analogs of methylumbelliferyl (MUF) (MUF-phosphate and MUF-ß-Glucopyranoside, Sigma Aldrich, St. Luis, MO, USA) were used, respectively. For determination of Leucine Aminopeptidase activity (LAP), L-leucine-4 Methyl-7-coumarinylamide (Leu-MCA, Sigma Aldrich, USA) was used. Five growing concentrations of each fluorogenic compound were added to water samples in three replicate. Incubation was carried out at the in situ temperature for 3 h. For Leu-MCA and MUF substrates, the fluorescence was measured with a Jenway 6280 fluorimeter, (380–440 and 365–445 nm filters respectively excitation-emission wavelengths). The increase of fluorescence was converted into the hydrolysis rate using a standard curve of MUF and MCA [13]. Vmax values (nanomol L^−1^ h^−1^) were converted into nanograms of C or P per liter and per hour, using a factor of 72 for the compounds with 6 C atoms and a factor of 31 for P [31].

Respiration rates were expressed as carbon dioxide production rate (CDPR) and were obtained from the O_2_ reduced by the electron transport system (ETS) activity. The Arrhenius equation (energy activation E_a_ 15.8 kcal mol^−1^) was adopted according to [32] to correct the rates for the difference between the in-situ temperature and the temperature at which the assay was performed. The determination of respiration via ETS relies on a conversion factor to convert the measured absorption of the assay into oxygen consumption or the carbon dioxide production rate [33]. ETS values (µL O_2_ L^−1^ h^−1^) were converted in carbon units as CDPR, (µg C L^−1^ h^−1^), applying the following equation:CDPR = ETS × (12/22.4) × (122/172)
where 12 is the C atomic weight, 22.4 the O_2_ molar volume and 172/122 the Takahashi oxygen/carbon molar ratio [32].

### 2.4. Statistical Analysis

A first univariate approach was used to assess significant differences in the values of the same variable in a different area (one-way ANOVA). Differences were considered significant when reaching at least a 0.05 probability level. In addition, Pearson correlation coefficients were calculated to highlight whether the prokaryotic activities have significant relationships with the physical-chemical characteristics of the water column (S, T, and OX), prokaryotic characteristics (PA and PB), and POM variables.

Multivariate analyses were also performed. In order to highlight similarities between groups of samplings, the Analysis of Similarities (ANOSIM) was carried out using the PRIMER 6β program package [34]. ANOSIM considers the differences between sample groups (defined a priori), using permutation/randomization methods on the resemblance matrix. We applied ANOSIM on similarity matrices (Euclidean distances), after data normalization.

Five matrices were created: (i) prokaryotic activity (using enzymatic activities and the CO_2_ production rates inferred by ETS evaluations), (ii) physical-chemical data (S, T, OX), (iii) prokaryotic variables (abundance and biomass), iv) POM concentrations (POC, PN, proteins, and carbohydrates), and (v) OM-related ratios (POC/PN, protein/carbohydrate, contribution of proteins and carbohydrates to POC).

These matrices were analyzed with SIMPER to highlight first differences among macro-areas (TNB and RS), and then among sub-areas (TNB-W, TNB-mid, TNB-E, RS-S, and RS-N).

The similarity percentage analysis (SIMPER, PRIMER 6β software version 6.1.5, 2006, PRIMER-E, Plymouth, UK) was applied to the normalized data of the two separated macro-areas. SIMPER was used to assess which factors affected variability and are primarily responsible for an observed difference. With this approach, the eventual dominance of one or more variables in the other variables distribution was highlighted.

Redundancy Analysis (RDA) [35], was performed on the normalized data of the two macro-areas (Brodgar version 2.7.5, 2017, Highland Statistics Ltd., www.highstat.com) to verify the influence of the bacterial and POM variables on the prokaryotic activity. RDA allows us to highlight the influence of the variation of one set of variables on the variation in another set of variables. Linear relationships between components of response variables (in this case those related to prokaryotic activity) that are explained by a set of explanatory variables (prokaryotic variables and POM) are summarized by RDA. A permutation test was applied (number of permutations: 499) to test whether the explained variation is larger than random contribution.

## 3. Results

The results of the studied variables are provided as mean values of 0–100-m layer (epipelagic) in the two macro-areas and as values of the samples collected below 200 m (mesopelagic) in the TNB macro-area (Table A1). The details on the average profiles of each sub-area for the epipelagic layer are reported in the following figures.

### 3.1. Environmental Parameters

The epipelagic layer (0–100 m) was mainly occupied by Antarctic surface waters (AASW) that, according to Orsi and Wiederwohl [36], are generally characterized by temperature values (T) ranging between −1.8 °C and 1.0 °C and salinity values (S) lower than 34.50. In the coastal TNB, the salinity value characteristic of AASW could change to 34.60, though maintaining an OX concentration of 11.2 ± 0.8 mg L^−1^ [5].

In TNB, the sudden changes of the vertical trends of T and S indicated a water column stratification at ca. 40–50 m depth (Figure 2). In RS, instead, the decrease of T and increase of S with depth were more gradual, generating a slighter and deeper stratification below 50–60 m depth. T mean values decreased with depth, more sharply in TNB macro-area than in RS. As shown by Figure 2A,D, related to the average T values of the different sub-areas, only TBN-mid and TNB-E showed 100-m-depth values slightly lower than −1.8 °C, indicating the mixing with cooler deep waters. S was generally lower than 34.5 in RS and 34.6 in TNB, confirming the presence of AASW (Figure 2B,E). Only some deep depths showed salinities slightly higher than the reference values, such as in RS-mid (100 m, 34.55 ± 0.04) and RS-S (100 m, 34.51 ± 0.01), and below 60 m in the TNB macro-area (34.62–34.66). Significant differences were found in the RS sub-areas, with RS-N showing significantly lower salinity than RS-S (one-way ANOVA, *p* < 0.05), and RS-mid (*p* < 0.05).

Mean OX concentration in the epipelagic layer (Figure 2C,F) showed significant differences within the macro-areas. Higher values in RS-S (one-way ANOVA, *p* < 0.05) and in RS-mid (*p* < 0.001) than in RS-N were found. TNB-W stations showed values significantly higher than TNB-mid (*p* < 0.01) and TNB-E (*p* < 0.001).

Below 200 m, T showed values ranging between −1.89 and −1.99 °C. Given that in stations 22 and 23 (TNB-E), T values were next to the freezing point, S values were 34.82, and OX concentration 9.6 ± 0.1 mg L^−1^, the presence of high salinity shelf waters (HSSW) was significant. HSSW is the most saline and the densest water that fill the bottom layer of the TNB area and, flowing northward, takes part in the formation of the Antarctic bottom water (AABW) [37]. Slightly lower S values for the other stations (34.72–34.77), and OX concentrations of 9.8 ± 0.4 mg L^−1^, indicated the mixing between HSSW and AASW. Potentially, other water types were present, such as the Terra Nova Bay ice shelf water (TISW), previously found in the 150–350 m layer and characterized by S values of about 34.70 [37,38,39].

### 3.2. Particulate Organic Matter

The vertical trends of the particulate organic matter (POM) are reported in Figure 3. The analyses showed variable specific standard deviations, depending on environmental variability that was highlighted by the analysis of the replicates. TNB showed an average specific standard deviation slightly higher than RS for PN and proteins (for both variables 16% in TNB vs. 11% in RS). POC showed similar average specific standard deviations (11% vs. 12% for TNB and RS, respectively), and carbohydrates 14% for TNB and 12% for RS.

The coastal-TNB showed, on average, higher concentrations of POC (168.6 ± 111.5 µg L^−1^) and PN (27.7 ± 20.1 µg L^−1^) than the offshore-RS (140.1 ± 57.1 µg L^−1^ and 20.1 ± 8.1 µg L^−1^, respectively), but lower values for proteins (148.0 ± 102.3 µg L^−1^ vs. 168.1 ± 67.8 µg L^−1^, for TNB and RS respectively) and carbohydrates (65.5 ± 39.1 µg L^−1^ vs. 92.2 ± 40.2 µg L^−1^, for TNB and RS respectively). These differences were significant for PN (one-way ANOVA, *p* < 0.05) and carbohydrates (*p* < 0.01). In the coastal TBN macro-area (Figure 3A–D), the concentrations showed a rather sharp decrease from the upper two depths (surface and intermediate 1) to the deeper intermediate 2 and 100 m. In the offshore RS, instead (Figure 3E–H), POC, PN, and protein values were similar down to the intermediate 2 and decreased at 100 m. In the southern sub-area (RS-S), protein values were similar in the entire water column. Carbohydrates were more variable, showing, on average, higher values in the RS-S than in the RS-N sub-area, especially below 20 m depth. The single station 43, placed in the middle between the coastal-TNB and the offshore-RS, showed the highest protein concentrations (227.5 ± 135.7 µg L^−1^), while the other variables showed values similar to the coastal-TNB macro-area.

### 3.3. Qualitative Ratios of POM

The qualitative POC/PN ratio was significantly (one-way ANOVA, *p* < 0.05) lower in the coastal TBN (6.6 ± 1.1) than in the offshore RS (7.2 ± 1.3). The trends of the different sub-areas were rather similar (Figure 4A,C). Slight increases with depth were observed. In the coastal TNB macro-area values higher than 8 were observed at 100 m in station 20 (8.8 ± 0.4) and 39 (9.8 ± 0.6), and in the anomalous surface depth of station 21 (8.5 ± 2.9). In the offshore RS, POC/PN ratio values higher than 8 were observed at 100-m depth of stations 59 (8.0 ± 0.3) and 57(9.3 ± 0.9). The single station 43 showed mean ratio values below 6 in the first three depths.

In the entire offshore RS, the average value for the protein/carbohydrate ratio was 2.2 ± 1.2, in the coastal TBN, it was 2.3 ± 0.9. A slight and irregular decrease was observed with depth, except for the offshore RS-N (Figure 4B,D). These stations showed the highest protein/carbohydrate ratio values in the intermediate depths and at 100 m. Similarly, the single station 43 showed ratio values next to 5 in the intermediate depths.

The contribution of the protein and carbohydrate carbon to the POC showed significant differences between the macro-areas (one-way ANOVA, *p* < 0.001) and the sub-areas. In the offshore RS-N, the protein contribution was significantly (*p* < 0.001) higher than in the RS-S, reaching 66 ± 8% vs. 50 ± 8%. In the TNB-W and TNB-mid stations, the contribution was lower (about 40%) while the TNB-E stations showed a significant (*p* < 0.001) increase up to 54 ± 13%. The single station 43 showed a protein contribution of 64 ± 15%, similar to the offshore RS-N stations. The carbohydrate contribution of TNB increased from TNB-W (15 ± 5%) to TNB-E (21 ± 5%) stations (significantly higher in the latter sub-area, *p* < 0.05). No significant differences were found for the RS-offshore (26 ± 10%). The single station 43 (RS-mid) showed values similar to the coastal TNB (17 ± 7%).

### 3.4. Microbiological Parameters in the Epipelagic Layer

In the epipelagic layer, a high variability of the microbial parameters was observed (Figure 5).

In TNB, PA (mean values for each sub-area are reported in Figure 5A,D) showed values ranging between 0.61 ± 0.28 × 10^6^ cell mL^−1^ (station 39) to 1.68 ± 0.12 × 10^6^ cell mL^−1^ (station 15, TNB-W). In the TNB area, higher abundances than the RS-offshore area were generally observed (one-way ANOVA, *p* < 0.01), where values ranged from 0.18 ± 0.07 × 10^6^ cell mL^−1^ (station 57, RS-N) to 1.06 ± 0.42 × 10^6^ cell mL^−1^ (station 46, RS-S). In the RS area, higher PA were found in the RS-S than in RS-N (*p* < 0.001) and RS-mid (*p* < 0.01), and RS-N showed values significantly lower than RS-mid (*p* < 0.001). In RS-S and RS-mid, peaks at the intermediate depths were also observed.

The TNB values of PB (Figure 5B) ranged between 8.7 ± 6.1 µg C L^−1^ (station 21, TNB-E) and 22.4 ± 3.7 µg C L^−1^ (station 15, TNB-W). The RS values (Figure 5E) ranged between 6.9 ± 4.4 µg C L^−1^ (station 57, RS-N) and 38.5 ± 15.0 µg C L^−1^ (station 46, RS-S). Higher PB was observed in RS-S than in RS-mid (*p* < 0.01) and RS-N (*p* < 0.001). PB showed trends for the different sub-areas that only partially resembled those of PA.

In the TNB, significantly higher values of CDPR than in RS (*p* < 0.001) were observed (mean values for each sub-area are reported in Figure 5C,F), despite they were rather variable and ranged from 0.07 ± 0.03 µg C L^−1^ h^−1^ (average for station 15, TNB-W) and 0.41 ± 0.06 µg C L^−1^ h^−1^ (station 18, TNB-mid). In the RS macro-area, the value ranged between 0.03 ± 0.01 µg C L^−1^ h^−1^ (station 43, RS-mid) and 0.36 ± 0.19 µg C L^−1^ h^−1^ (station 80, RS-N). No significant differences were observed in the TNB sub-areas. Particularly low values in RS-mid and RS-S sub-areas were found, while RS-N sub-area showed significantly higher respiration rates (*p* < 0.001 for RS-S and *p* < 0.001 for RS-mid), similar to the TNB area.

LAP activity (mean values for each sub-area are reported in Figure 6A,E) ranged between 9.1 ± 9.0 ng C L^−1^ h^−1^ (average for station 13, TNB-W) and 222.3 ± 111.0 ng C L^−1^ h^−1^ (station 46, RS-S). A decreasing trend with depth was observed and slightly higher values characterized TNB-W. The RS area showed significantly higher mean values than TNB (*p* < 0.001), with strong variations with depth. LAP above 50 m was approximately 3 times higher than below 50 m depth (271.7 ± 90.6 ng C L^−1^ h^−1^ and 89.5 ± 78.3 ng C L^−1^ h^−1^, respectively for the upper and lower layer). The same ratio was observed also for the TNB macro-area (58.0 ± 32.3 ng C L^−1^ h^−1^ and 20.4 ± 21.5 ng C L^−1^ h^−1^, respectively for the upper and lower layer).

GLU activity showed a wide range of values in the epipelagic layer and significant differences among the macro-areas (RS vs. TNB, *p* < 0.01) (mean values for each sub-area are reported in Figure 6B,F). In TNB, the values ranged between 41.7 ± 10.7 ng C L^−1^ h^−1^ (station 19, TNB-mid) and 197.4 ± 56.4 ng C L^−1^ h^−1^ (station 15 and 39, TNB-W). In RS, the values ranged between 104.8 ± 71.8 ng C L^−1^ h^−1^ (station 59, the northernmost of RS-N) and 321.5 ± 111.2 ng C L^−1^ h^−1^ (station 46, RS-S). In TNB, high values in TNB W than TNB-mid and TNB-E stations were found (*p* < 0.001). In the RS macro-area, RS-N GLU was significantly lower than RS-S (*p* < 0.01), and significantly lower values in the RS-mid than RS-N (*p*< 0.004) and RS-S (*p*< 0.01) were found.

In the epipelagic layer of TNB, AP (mean values for each sub-area are reported in Figure 6C,G) showed trends similar to those of GLU. A significant difference was recorded between TNB and RS (*p* < 0.01). In TNB, the TNB-W showed significantly higher values than TNB-mid (*p* < 0.05) and TBN-E (*p* < 0.05). The mean values ranged between 11.3 ± 9.2 (station 18, TNB-mid) and the exceptional 244.7 ± 155.7 ng P L^−1^ h^−1^ (station 81, TNB-W). In RS, the values ranged between 58.5 ± 46.2 ng P L^−1^ h^−1^ (station 59, RS-N) and 151.2 ± 108.0 ng P L^−1^ h^−1^ (station 43, RS-mid). RS-mid AP was significantly higher than RS-N (*p* < 0.05).

A prevalence of GLU on LAP was generally observed with LAP/GLU ratio <1 in most of the sub-areas and depth of TNB, where the LAP/GLU ratio ranged between 0.01 and 2.21 (Figure 6D). Significantly higher ratios (*p* < 0.001) were found in RS area (ratios between 0.06−3.52) (Figure 6H). In particular, LAP/GLU ratios were >1 at surface and sub-surface layers; an inversion of LAP/GLU ratio was observed at 60–100 m depths.

### 3.5. Microbiological Parameters in the Sub-Euphotic Zone

The data collected below the 200 m in the TNB (Table A1), highlighted that the microbial abundance and activities declined differently.

In particular, deep PA showed significantly lower values than the epipelagic layer (one-way ANOVA, *p* < 0.05), decreasing from 1.11 ± 0.37 × 10^6^ cell mL^−1^ to 0.40 ± 0.54 × 10^6^ cell mL^−1^, although station 13, 15 and 20 maintained similar abundances. The PB showed the same decreasing trend with the exception of station 15.

Significant (one-way ANOVA, *p* < 0.01) decreases down to 2% were showed by CDPR below 200 m, reaching values from 0.283 ± 0.150 µg C L^−1^h^−1^ of the epipelagic layer to 0.005 ± 0.002 µg C L^−1^ h^−1^ of the deep layer. A significant decrease (one-way ANOVA, *p* < 0.01) of the LAP activity was also observed, on average from 39.1 ± 21.8 ng C L^−1^ h^−1^ of the epipelagic layer to 7.0 ± 7.3 ng C L^−1^h^−1^ of the deep layer. Only the 400 m deep value of station 13 showed an increase of 147%. GLU and AP, instead, showed different trends, with several stations that maintained the epipelagic activities. The stations 20, 22, and 81 increased AP rates from 160% to 165%, and GLU rates from 114% to 158%.

### 3.6. Relationships of OM with the Enzymatic Variables

Protein and carbohydrate turnover times (Figure 7A,C for proteins and Figure 7B,D for carbohydrates) showed significantly lower values in the offshore RS than in the coastal TNB (one-way ANOVA, protein turnover time: *p* < 0.001, carbohydrate turnover time: *p* < 0.05), especially at the shallower depths (on average 47.7 ± 60.7 d for proteins and 10.3 ± 5.6 d for carbohydrates in RS vs. 280.7 ± 593.1 d for proteins and 16.4 ± 13.5 d for carbohydrates in TNB). In both macro-areas, proteins showed higher turnover times than carbohydrates (one-way ANOVA, *p* < 0.05 for TNB, *p* < 0.01 for RS), due to the lower protein hydrolysis activity (LAP) than the glucosidase one (GLU). The trends of the turnover times were rather similar in the offshore RS for both the variables. In the coastal TNB, the values were irregular for proteins, while the carbohydrate turnover time tended to decrease more regularly with depth.

### 3.7. Correlation Analysis and Multivariate Analyses

The prokaryotic activities showed different correlations among them in the two macro-areas (Table 2). CDPR did not show significant correlations, except in the offshore RS, where it was significantly and negatively correlated to GLU and PA. More frequent were the correlations among the enzymatic activities. In TNB, the three enzymatic activities correlated, while in RS the AP did not correlate with GLU. A general decoupling between prokaryotic activities and PA or PB was observed, with few exceptions, such as GLU and PA in RS.

The correlations between prokaryotic activities and the other variables (physical-chemical, and POM) showed that CDPR did not depend on the physical-chemical characteristics of the water column, nor the quantity/quality of POM. The only exception was a negative correlation with S in RS. The enzymatic activities were more related to OM, especially LAP that in both the macro-areas was mostly expressed in case of high OM availability and, in the coastal TNB, in case of good quality of OM (namely lower POC/PN ratio and higher protein/carbohydrate ratio). It is worth noting that LAP correlated also with the physical-chemical characteristics of the water column, due to its strong link with POM, while AP and particularly GLU showed a lower number of significant correlations both for the physic-chemical characteristics and the POM variables. In the TNB all the POM variables showed significant negative correlations with salinity (for instance POC: *r* = −0.79, *n* = 36, *p* < 0.01) and significant positive correlations with temperature (for instance POC: *r* = 0.93, *n* = 36, *p* < 0.01). In RS, instead, POM showed some significant negative correlations with salinity (for instance POC: *r* = −0.45, *n* = 32, *p* < 0.05, and also PN and proteins), but carbohydrates showed no significant correlation.

A multivariate SIMPER analysis (Table 3) was applied separately on the two macro-areas, to highlight the main variables that contribute to the variability within each macro-area.

In the coastal TNB, it is clear a pivotal role of the physical-chemical variables T and S (explaining 10.4% and 10.0%, respectively, of the dissimilarity), but also of the bulk POM, as PN (10.4%) and POC (10.1%) and of the enzymatic activity devoted to remineralization (AP, 10.1%), explaining together 51.9%. In the offshore RS, the highest value was observed for OX (12.6%), followed PB (11.7%) and LAP (11.3%). Moreover, the qualitative ratios of POM (POC/PN ratio and protein/carbohydrate ratio) contribute to reach 53.6% of explained dissimilarity (Table 3).

Multivariate ANOSIM was applied to five groups of variables (see Material and Methods for details), highlighting differences among the two macro-areas (Table 4) especially for the prokaryotic activity and the OM ratios.

A more detailed analysis related to the different sub-areas (Table 5) for what concerns the prokaryotic activity showed that within the TNB sub-areas, the TNB-E was not different from the others, but TNB-W and TNB-mid showed a higher difference. TNB sub-areas were different from the RS sub-areas (except for TNB-E vs. RS-N). The RS sub-areas were different.

The differences observed for the prokaryotic activity did not depend always on differences of the prokaryotic variables (abundance and biomass), but rely also on POM quantity and quality (Table 5). In particular, the POM quantitative variables highlighted differences between the sub-areas belonging to the coastal TNB and those belonging to the offshore RS, while a general similarity was recorded inside the macro-areas.

The multivariate RDA results (Table 6) for the TBN coastal macro-area showed that the sum of all canonical eigenvalues was 0.55 (axis 1:0.36, axis 2:0.11), for the offshore RS macro-area the sum was 0.64 (axis 1:0.34, axis 2:0.22). The costal TBN prokaryotic activity was significantly influenced by the quality of the POM (namely POC/PN ratio) and by the quantity of nitrogen-containing POM (PN). Salinity was the physical constraint that exerted a control, as well.

In the offshore RS, the POM bulk (POC), the biochemical fraction of carbohydrates, and the POC/PN ratio achieved a major role in the prokaryotic activity regulation, together with the prokaryotic abundance. Salinity, as in the coastal TNB, showed a significant role.

## 4. Discussion

### 4.1. General Characteristics of the Studied Area

Our data indicated that, in summer 2017, the two macro-areas showed significant differences in the distribution of OM and in some physical-chemical characteristics that contributed to regulate the prokaryotic activity rates.

The multivariate ANOSIM revealed significant differences in terms of physical-chemical characteristics of the epipelagic layer, despite both macro-areas showed the dominance of AASW. The ANOSIM applied on all the sub-areas showed that the differences were minor inside the TNB macro-area, that in previous studies was divided by a frontal structure that had a role in the vertical distribution of POM and phytoplankton biomass [23]. In the summer of 2017, the three geographical sub-areas were rather homogeneous and showed T and S values opposite to those found previously. The coastal waters (TNB-W) were warmer than the others and saltier than the eastern stations (TNB-E). In the previous study, the frontal structure was due to the convergence of ice-influenced coastal waters and ice-free offshore waters [5], in the present study the ice influence was not revealed, leaving the TNB-W stations to develop in a full-summer scenario. Moreover, the physical-chemical characteristics of the RS sub-areas were different, with the RS-N significantly fresher than RS-S and RS-mid. This difference may be due to the general seasonal features of the RS, where the Ross Ice Shelf winter polynya starts to enlarge in early spring in the southern area of RS, decreasing in this zone the influence of meltwater by advection [21]. On the contrary, in the area north of 75°S meltwater endures for longer times [2], thus explaining our lower S values.

The OX concentration in the upper layers of the water column is related to continuous exchanges with the atmosphere, but also to the presence of active photosynthetic organisms [4]. In this case, despite the negligible physical differences within the TNB sub-areas (ANOSIM), the significantly higher OX values of TNB-W than TNB-mid and TNB-E suggest a high phytoplankton activity in the coastward stations, that may influence the OM availability and, in turn, the prokaryotic activity [1,16,40]. The significantly higher OX values of RS-S and RS-mid distinguish between an interior-RS productive ecosystem vs. an RS-N ecosystem with the tendency to oligotrophy. This distinction is in accordance with [41], who reported a marked optical front that separates the high-biomass southern RS from the low-biomass northern area. In the scenario described by [41], our RS-N stations fall in an intermediate stripe.

Phytoplankton distribution and typology depend on the water column structure, for instance on the presence and intensity of density gradients that differently favor the main phytoplankton taxa (diatoms or *Phaeocystis antarctica*) [3]. In addition, the distribution of the POM in the water column depends, mainly, on the phytoplanktonic production, being land-inputs virtually absent in the RS [7,42].

The vertical distribution of POM in our study indicated a variability of the processes influencing POM availability to prokaryotic activity, depending on POM origin and the physical modifications of the water column. Strong differences between the upper epipelagic layer and the lower epipelagic one were observed in TNB, while a higher homogeneity was found for RS. Physical stratification forced the OM to accumulate in the shallower layers in the coastal TNB; in the offshore RS, seasonal and meteorological conditions forced POM to be spread in a thicker layer, down to 50–60 m. The RS characteristics were probably reinforced by the sudden bad sea conditions that occurred between the sampling of the RS-S and the RS-N sub-areas, with wind and wave mixing action that may have homogenized the surface water layers down to the second intermediate depth.

### 4.2. General Characteristics of the POM and Its Relationship with Prokaryotic Variables

The role of the physical-chemical variables within an area was highlighted also by the multivariate SIMPER analysis (Table 3), that however indicated that also POM characteristics were among the leading variables that differently shaped the two macro-areas. In TNB, in fact, both T and S influence deeply the distribution of the other variables, but the bulk POM (POC and PN) played a similar role. In the RS, instead, the main forcing was OX, indicating that processes related to OX liberation (namely, primary production and therefore the quality of POM) were the reference scenario that all the considered variables had to deal with. It is the case of PB, and of the degradative enzymes (especially LAP).

The results suggest that in TBN accumulation of POM prevails, leading to an increase of the refractory fraction, given that the percentage contribution of the main biochemical components (the labile proteins and semi-labile carbohydrates) to POC was significantly lower in coastal TNB than in offshore RS. This accumulation deals with the seasonal development of the productive processes. During spring, high primary production could be influenced by land ice melting, that increases the inorganic nutrient concentrations and leads to massive phytoplankton development [3]. A previous series of blooms during spring could have led to the POM accumulation. The heterotrophic consumption of the more labile fraction of OM and/or its physical aging [22], were responsible for the increase of the refractory (non-protein or carbohydrate) part of POM.

The relevant role of AP in the SIMPER analysis for the TNB macro-area indicated that this enzyme had not only the function of providing P to the micro-organisms (in this case the higher AP values of TNB-W than the other TNB sub-areas could be due also to expression by autotrophs [43]). AP could be used to detach P from molecules such as proteins, allowing other enzymes such as LAP to perform degradation [44]. In TNB, the three hydrolytic enzymes were, in fact, highly correlated to each other and their activity increased in case of good trophic quality POM, as indicated by the correlations with the qualitative ratios. The RDA indicated the role of proteins and PN in the prokaryotic activity regulation, confirming that nitrogenous compounds are a pivotal food source for prokaryotic organisms [45].

In RS-mid, the particularly high values of AP, often exceeding LAP activity at all depth (on average LAP/AP ratio was 0.83 ± 0.59), were probably related to a major need of P for phytoplankton in the euphotic layer [40].

As in TNB-W, the higher OX concentrations of RS-mid were in accordance with this hypothesis, denoting high productivity. In addition, AP may be produced both by prokaryotes (bacteria and cyanobacteria) and eukaryotes (phyto- and zoo-plankton). Other organisms possess phosphoester hydrolases (phosphatases), able to release inorganic phosphorus for metabolic needs [46]. It has been demonstrated that phagotrophic nanofagellates and marine invertebrates might release extracellular enzymes through major physiological processes such as excretion or sloppy feeding. Recently, active extracellular phosphatase was found in the feces of crustacean species; these can substantially contribute to the pool of active extracellular phosphatases and to the recycling of phosphorus in aquatic systems [47].

In TNB, higher values of CDPR and lower of EEAs than in RS could suggest that the coastal area acted mainly as a “sink,” according to Williams [48], who proposed this model for the DOM. “Sink” means mainly “respiring OM”. In oceanic waters, [48] have shown that respiration increases as a response of DOM production in case of intense phytoplankton bloom events. In our study, the absence of correlations between CDPR and POM could depend on the role of DOM in regulating the respiration prokaryotic activity. The distribution of plankton degradation of organic matter is often related to the different hydrological and trophic conditions [8].

On the other hand, the offshore RS was characterized by higher concentrations of labile and semi-labile compounds, that provide a valuable POM source to microbial development. The higher values of the enzymatic activities in the offshore area than in TNB suggested the occurrence of primary production events. The addition of fresh labile OM sources could have stimulated fast-growing micro-organisms that have high P demand. In fact, [3] reported increasing exoenzyme activity associated with phytoplankton blooms in the Amundsen Sea, Antarctica.

The RDA highlighted that in case of high concentrations of freshly produced OM, such as in the offshore RS, the prokaryotic activity was significantly regulated by this POM and by the abundance of the prokaryotic organisms. Thus, microbes quickly adapt to the environmental characteristics to achieve a higher gain from the degradation and remineralization processes. The high enzymatic activities of the offshore RS and the lower CDPR especially in the RS-S sub-area, indicate that in that moment, the prokaryotic organisms acted in the trophic chain as a “link”, i.e., recycling and transferring organic material to higher trophic levels [48].

In the same summer period, an unusual bloom of bacteriophagous cells, belonging to loricate choanoflagellates, was observed by [49], in the RS-mid sub-area. The presence of peculiar planktonic components was also revealed by the POM characteristics. Despite POM concentrations were similar to the TNB ones, the station 43 of the RS-mid showed particularly high protein concentrations, in accordance with the proteic composition of the choanoflagellate lorica [50]. The high protein concentrations led to low POC/PN ratio values and increased notably the protein/carbohydrate ratio values, indicating the presence of high trophic value POM. In this area, the choanoflagellates may control the abundance and biomass of prokaryotes and the increased transfer of materials and energy to the high trophic levels, growing the strength of the microbial loop.

### 4.3. Peculiar EAA Expression Related to POM Composition

In the coastal TNB, the turnover times of proteins and carbohydrates were significantly higher than in the offshore RS, confirming that a part of these components was not easily degraded and endured in the system. It is worth noting that the turnover times of carbohydrates were lower than those of proteins, suggesting that the chemical structure of the glucidic substances was not a limiting factor for prokaryotes and that these substances can be easily used.

Especially in the RS macro-area, a major role of carbohydrates was confirmed by the RDA and by the significant correlations between PA and carbohydrates and with GLU, meaning that the micro-organisms were adapted to the exploitation of semi-labile substances [3,40]. They, likely, belong to the group of opportunistic micro-organisms that flourish in high OM concentrations [7]. These authors observed that high concentrations, especially of freshly produced OM such as the proteins and carbohydrates found in the RS macro-area, favored fast-growing copiotrophs that outcompeted slow-growing oligotrophic micro-organisms.

Carbohydrates can be relevant for microbial metabolism in the ocean since they are major constituents of phytoplankton and of marine particles from marine snow to large aggregates [16]; moreover, the GLU completes the hydrolysis of cellulose in marine particles [51]. In Arctic Surface Waters [14] and in western Antarctica [52], other Authors showed that bacterioplankton used preferentially carbohydrates, and the glycosidase assayed (i.e., beta-glucosidase, xylosidase, arabinosidase, and cellobiosidase) had high specific activities. In those studies, the influence of polar ice-melting was indicated as a leading factor. Ice-melting increased the relative importance of polysaccharides in the water column, releasing the organic materials embedded in the ice structure, thus affecting the spectra of bacterial extracellular enzymes [11,14]. But, due to the lack of ice in our study area, this cannot be the main reason for the shorter carbohydrate turnover times we found.

Previous studies performed in the Ross Sea have shown that a dominance of particulate carbohydrates depended on the development of *P. antarctica* [23]. In the Arctic sea, [45] observed that the prokaryotes were linked to *P. pouchetii* blooms, being this organism especially active in regenerated conditions, when the competition with diatoms was relaxed. Therefore, the surprisingly low turnover times for carbohydrates may be related to the labile POM produced by *P. antarctica* blooms.

In many marine environments, proteins are often processed faster than polysaccharides, as showed by LAP to GLU ratios being >1 [24,31,46]. On the contrary, in our study, a balance between the two activities or a prevalence of GLU on LAP were observed, with LAP/GLU ratio < 1. Considering these molar ratios as an indicator of microbial metabolism of organic C and N, C was preferentially mobilized compared to N due to high abundance of digestible carbohydrates [46].

The LAP/GLU ratio in our study was different from the ranges observed by Sala et al. [14] in the Arctic ocean (85–100), and from temperate regions (as in the oligotrophic Mediterranean Sea) where LAP prevail over other enzymatic activities [13,46]. Nevertheless, a prevalence of GLU on LAP was observed in Arctic fjords [20], and this suggests that patterns of peptide hydrolysis may differ between temperate and high latitude sites [52]. A higher hydrolysis potential for polysaccharides relative to proteins in Arctic Ocean, was showed [45], reflecting changes in OM structure between the water masses.

In the RS, a changing ratio between the production of diatoms and of *P. antarctica* has been observed in the last years [4]. The peculiar GLU activity that was recorded in the summer of 2017 could be linked to these changes, indicating an efficient acclimation of prokaryotes to new POM quality characteristics.

### 4.4. Microbial Activities in Sub-Euphotic Zone

Although few data were collected below the 100 m in the TNB, they give some indication of the abrupt reduction of prokaryotes both as abundance and activity. Reduction of abundance as well as activities with depth was a common phenomenon over the world [9,53,54]. In the TNB this was likely linked to POM trophic quality. In fact, a decrease of the trophic quality of POM (higher POC/PN ratio values) was observed in some stations already starting from the 100-m-depth, and also reported in previous studies [23].

However, hot spots of prokaryotic activity and abundance were reported in some marine environments in the presence of aggregates, which provide a source of substrate for organisms at depth [9,10,44]. Misic et al. [22], found that CDPR increases in the subsurface layer of the TNB area were related to the need for additional energy by prokaryotic organisms to use low-quality POM. High AP also at greater depth was observed by [44], related to changes in available POM. In our study, spot-increases of AP and GLU occurred, with LAP/GLU ratios < 0.01 in several stations, probably related to aggregates of dead phytoplankton. Bacterioplankton diversity increased in the mesopelagic zone, probably supported by particles sinking out of the euphotic zone [7]; these local factors increased organic and inorganic nutrients.

In addition, in TNB, the chemical-physical characteristics (S was >34.72 and T between −1.89–1.99 freezing temperature; low OX content 9.2–9.7), indicated the presence of water masses other than the surface ones, namely TISW, and signals of HSSW [37,38]. In the entire RS, the prokaryotic activities in the deep water masses were variable and often rather high, depending on the origin of the water masses themselves [10,55]. Therefore, the occurrence of the hot-spots of activity could be linked to the intrusion and mixing of different water masses. Azzaro et al. [19] suggested that the amplitude of the vertical flux of the biogenic material in the Ross Sea was associated with summer phytoplankton blooms, as well as with the formation of different deep water masses, that potentially carry out pre-formed organic carbon load.

## 5. Conclusions

Autochthonous cold-adapted prokaryotic communities, thanks to their peculiar metabolic abilities, play a central role in the functioning of the Antarctic marine environment, recycling POM and supporting life cycles under extreme conditions. The transformation and remineralization of complex organic matter by marine heterotrophs vary spatially and temporally, regulated by different forcing.

In the context of climate change, our study makes a contribution to the mechanisms underlying the prokaryotic utilization of organic particles in the epipelagic ocean. We furnished evidence that environmental factors affect the composition and quality of POM, produced by phytoplankton bloom, and as a consequence, microbial communities in a cold environment.

Variations in both T and S, producing changes in water masses, lead to changes in the distribution of the prokaryotic variables, which differ regionally. In TBN, accumulation of POM prevails, leading to an increase of the refractory fraction with high turnover times of proteins and carbohydrates. LAP/GLU ratios show a higher hydrolysis potential for polysaccharides than for proteins in a major measure in TNB than in RS. These ratios seem to be suitable markers to describe the variability of area, thus reflecting the differences in organic matter composition between the areas.

In the offshore RS, higher concentrations of freshly produced POM (indicated by processes related to OX liberation), is observed; high AP also sustains productivity. The prokaryotic enzymatic activity increases and is regulated by the quality of POM and by the prokaryotic abundance. Proteins and carbohydrates are metabolized rapidly in this area (as shown by the shorter turnover times), due to enzymatic activities (LAP, GLU).

## Figures and Tables

**Figure 1 microorganisms-08-01273-f001:**
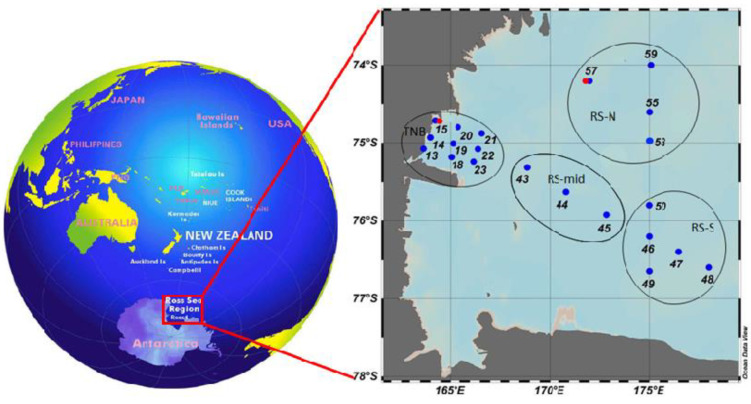
Study area in the Ross Sea (RS) and sampling stations. The macro-areas of Terra Nova Bay and RS offshore area were represented (in blue the sampling stations, in red the repeated stations).

**Figure 2 microorganisms-08-01273-f002:**
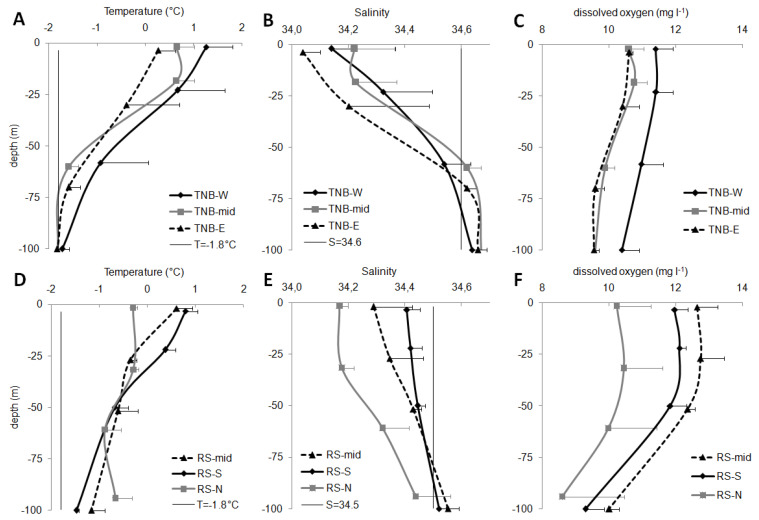
Physical and chemical variables of the epipelagic layer. Average values (+sd) for the two macro-areas (coastal TNB, upper panels, and offshore RS, lower panels), see text for details related to the sub-areas. (**A**,**D**): temperature (°C), the thin vertical line denotes the T value (−1.8 °C) indicated as the lower limit for Antarctic surface waters (AASW) [36]; (**B**,**E**): salinity, the thin vertical line denotes the S value indicated as the higher limit for surface waters in TNB (34.60) [5] and in RS (34.50) [36]; (**C**,**F**): dissolved oxygen concentration (mg L^−1^). Many depths were measured for physical-chemical parameters, we only reported four points for similarity with other figures.

**Figure 3 microorganisms-08-01273-f003:**
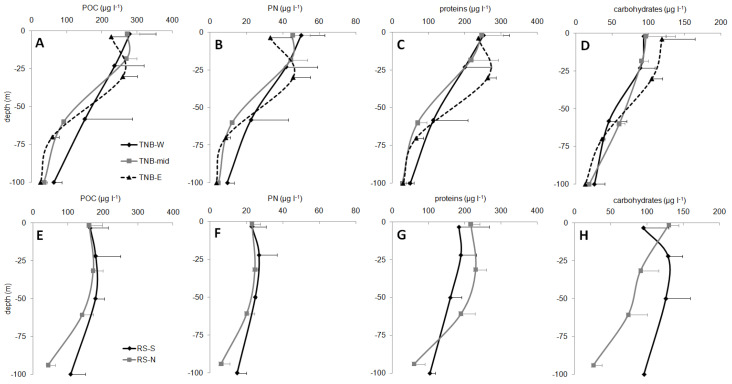
Concentrations of the POM (µg L^−1^) in the coastal TNB macro-area (upper panels (**A**–**D**)) and in the RS offshore macro-area (lower panels (**E**–**H**)).

**Figure 4 microorganisms-08-01273-f004:**
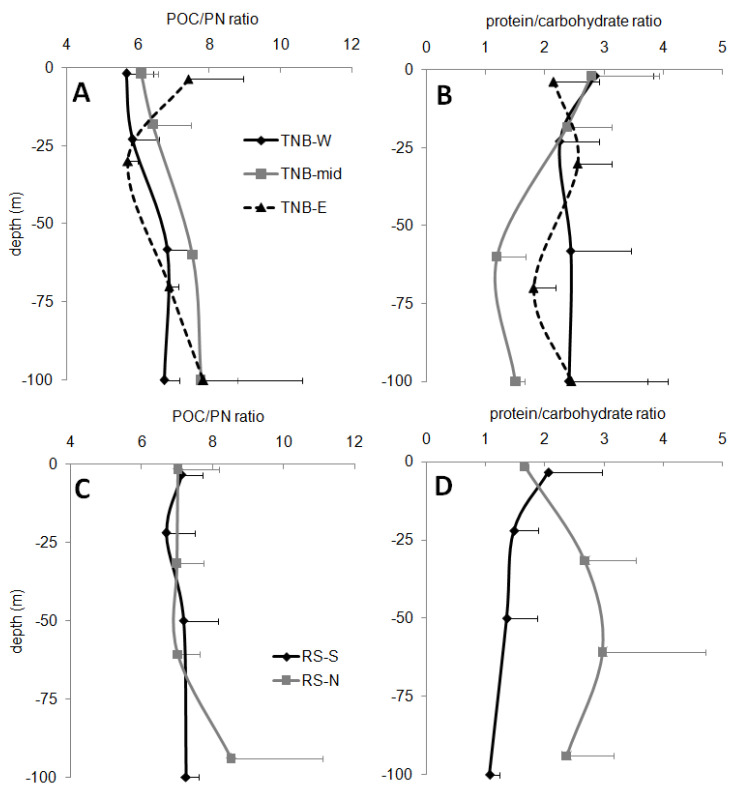
POM related ratios as POC/PN ratio and protein/carbohydrate ratio in the coastal TNB macro-area (upper panels (**A**,**B**)) and in the RS offshore macro-area (lower panels (**C**,**D**)).

**Figure 5 microorganisms-08-01273-f005:**
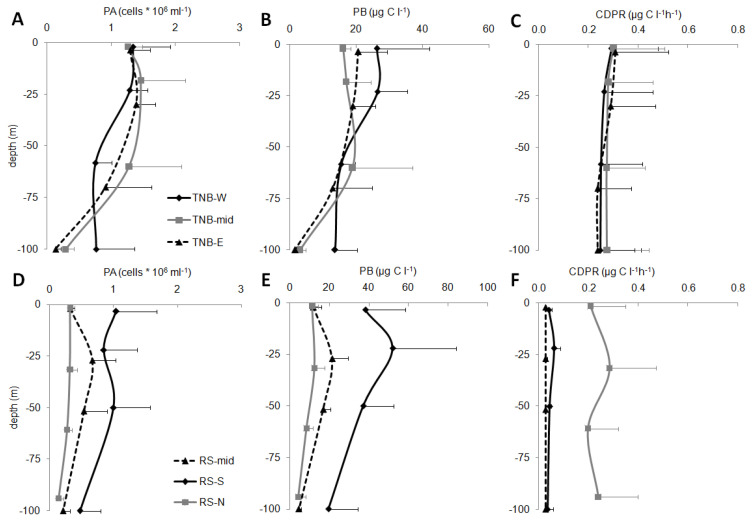
Vertical distribution of microbial parameters: PA (prokaryotic abundance, cells * 10^6^ mL^−1^), PB prokaryotic biomass (µg C L^−1^), and CDPR (carbon dioxide production rates, µg C L^−1^ h^−1^) in coastal TNB (upper panels (**A**–**C**)) and offshore RS (lower panels (**D**–**F**)).

**Figure 6 microorganisms-08-01273-f006:**
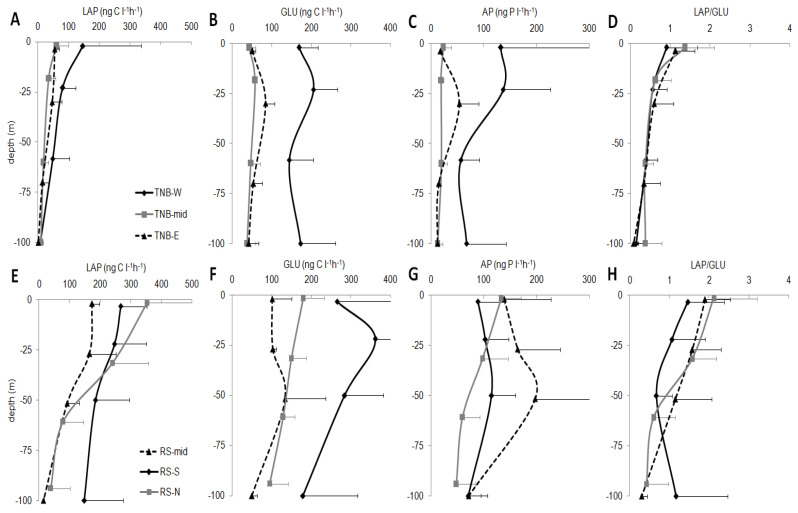
Vertical distribution of enzymatic activities: leucine-aminopeptidase (LAP, ng C L^−1^ h^−1^), β-glucosidase (GLU, ng C L^−1^ h^−1^), alkaline phosphatase (AP, ng P L^−1^ h^−1^), aminopeptidase/β-glucosidase ratio (LAP/GLU) in coastal TNB (upper panels, (**A**–**D**)) and offshore RS (lower panels, (**E**–**H**)).

**Figure 7 microorganisms-08-01273-f007:**
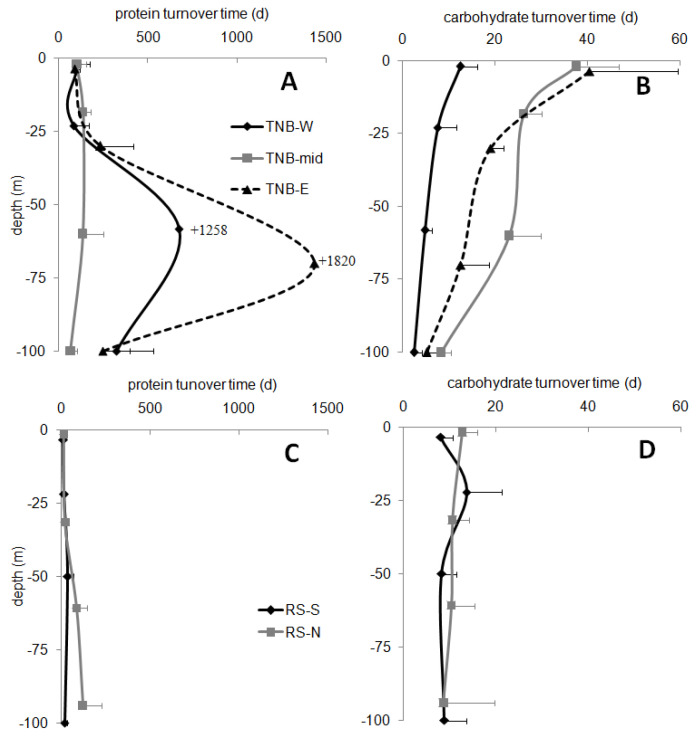
Turnover times of proteins and carbohydrates in the coastal-TNB macro-area (upper panels **A**,**B**) and in the RS-offshore macro-area (lower panels **C**,**D**).

**Table 1 microorganisms-08-01273-t001:** Position and depth of the sampled stations and their grouping in the main macro-areas and sub-areas.

Macro-Area	Sub-Area	Station	Date	Lat (°S)	Long (°E)	Depth (m)	n° Sampling Depths	Sampling for POM
TNB	TNB-W	13	13-gen	75.07	163.66	867	6	x
		14	13-gen	75.93	164.00	342	5	-
		15	13-gen	74.71	164.23	498	5	x
		39 (15 r)	22-gen	74.71	164.22	497	5	x
		81 (15 r)	11-feb	74.71	164.22	490	5	x
	TNB-mid	18	14-gen	75.18	165.04	1054	4	x
		19	14-gen	75.00	165.13	925	5	x
		20	14-gen	74.80	165.40	662	5	x
	TNB-E	21	14-gen	74.87	166.58	886	4	x
		22	15-gen	74.08	166.40	855	5	-
		23	15-gen	75.24	166.18	852	5	x
RS	RS-mid	43	25-gen	75.31	168.89	351	4	x
		44	25-gen	75.63	170.85	568	4	-
		45	25-gen	75.92	172.83	571	4	-
	RS-S	46	26-gen	76.20	175.00	568	4	x
		47	26-gen	76.40	176.50	409	4	-
		48	26-gen	75.60	178.00	292	4	-
		49	26-gen	76.65	175.00	425	4	x
		50	26-gen	75.80	175.00	403	4	x
	RS-N	53	28-gen	74.94	175.00	326	4	x
		55	29-gen	74.60	175.01	439	4	x
		57	29-gen	74.20	172.50	410	4	x
		59	30-gen	74.00	175.09	579	4	x
		80 (57 r)	10-feb	74.20	172.00	406	4	x

The number of 4 samplings was in the epipelagic layer (surface, intermediate 1 and 2, 100 m, see text for details), when higher than 4, the station was sampled also in the mesopelagic layer down to a maximum of 820 m (station 23). About sampling for POM, x indicated sampled station, -no sampled station.

**Table 2 microorganisms-08-01273-t002:** Significant correlations (bold numbers: *p* < 0.01, thin numbers: *p* < 0.05, ns: not significant) between the prokaryotic characteristics and activities (PA, PB, CDPR, LAP, GLU and AP) and the other variables, grouped for: physical-chemical (S, T, OX), POM quantitative (POC, PN, proteins and carbohydrates), POM qualitative (POC/PN ratio and protein/carbohydrate ratio). Correlations among the prokaryotic activity variables are also provided. Coastal TBN: *n*= 36, except for PA, PB, and GLU (*n* = 32). Offshore RS: *n* = 36.

Macro-Area		Physical-Chemical			POM Quantitative				POM Qualitative		Prokaryotic Activity			
	Variable	S	T	OX	POC	PN	Proteins	Carbohydrates	POC/PN	Prot/Carbo	CDPR	LAP	GLU	AP
TNB	PA	**−0.48**	**0.55**	**0.48**	**0.67**	**0.60**	**0.58**	**0.64**	ns	ns	ns	ns	ns	ns
	PB	−0.44	**0.48**	**0.46**	**0.53**	**0.49**	0.43	**0.61**	ns	ns	ns	0.37	ns	ns
	CDPR	ns	ns	ns	ns	ns	ns	ns	ns	ns	-	-	-	-
	LAP	**−0.51**	**0.57**	**0.55**	**0.61**	**0.67**	**0.69**	**0.49**	**−0.56**	0.42	ns	-	-	-
	GLU	ns	ns	0.38	ns	ns	ns	ns	−0.38	ns	ns	**0.41**	-	-
	AP	ns	ns	**0.58**	ns	0.37	0.35	ns	**−0.48**	0.36	ns	**0.55**	**0.50**	-
RS	PA	ns	ns	**0.52**	ns	ns	ns	**0.49**	ns	−0.41	**−0.37**	ns	**0.70**	ns
	PB	ns	0.42	**0.58**	**0.37**	0.36	ns	**0.47**	ns	−0.35	ns	ns	ns	ns
	CDPR	**−0.40**	ns	ns	ns	ns	ns	ns	ns	ns	-	-	-	-
	LAP	**−0.53**	**0.44**	**0.49**	**0.60**	**0.63**	**0.57**	**0.55**	ns	ns	ns	-	-	-
	GLU	ns	ns	**0.37**	0.42	ns	ns	0.40	ns	ns	**−0.39**	**0.37**	-	-
	AP	**−0.53**	ns	**0.40**	**0.55**	**0.54**	**0.49**	**0.58**	ns	ns	ns	**0.62**	ns	-

**Table 3 microorganisms-08-01273-t003:** SIMPER analysis results: main factors contributing to the grouping of the observations within the two macro-areas. (In bold the most relevant variables affecting the dissimilarity)

Variables	TNB Macro-Area			Variables	RS Macro-Area		
	Av.Sq.Dist	Sq.Dist/SD	Contrib%		Av.Sq.Dist	Sq.Dist/SD	Contrib%
LAP	0.10	0.48	0.7	PN	0.26	0.45	2.2
OX	0.34	0.51	2.2	PA	0.36	0.38	3.1
GLU	0.65	0.46	4.2	AP	0.40	0.52	3.4
PB	0.73	0.46	4.8	S	0.40	0.54	3.4
prot./carbo. ratio	0.77	0.49	5.0	T	0.40	0.47	3.4
POC/PN ratio	0.78	0.42	5.1	POC	0.40	0.47	3.5
CDPR	0.88	0.54	5.7	proteins	0.55	0.50	4.8
carbohydrates	0.88	0.50	5.8	CDPR	0.84	0.46	7.2
PA	1.13	0.48	7.4	GLU	0.88	0.39	7.5
proteins	1.26	0.54	8.2	carbohydrates	0.94	0.49	8.0
**S**	**1.54**	**0.51**	**10.0**	**prot./carbo. ratio**	**0.99**	**0.32**	**8.5**
**POC**	**1.54**	**0.54**	**10.1**	**POC/PN ratio**	**1.11**	**0.31**	**9.5**
**AP**	**1.55**	**0.27**	**10.1**	**LAP**	**1.32**	**0.50**	**11.3**
**PN**	**1.60**	**0.54**	**10.4**	**PB**	**1.37**	**0.38**	**11.7**
**T**	**1.60**	**0.54**	**10.4**	**OX**	**1.47**	**0.50**	**12.6**

**Table 4 microorganisms-08-01273-t004:** Results for the ANOSIM applied on the normalized and resembled (Euclidean distances) data for the macro-areas TNB and RS.

		Prokaryotic Activity		Physical-Chemical Characteristics		Prokariotic Abund. and Biom.		OM		OM Ratios	
		R Statistic	Sign. %	R Statistic	Sign. %	R Statistic	Sign. %	R Statistic	Sign. %	R Statistic	Sign. %
TNB vs. RS		0.25	0.1	0.17	0.1	0.19	0.1	0.15	0.1	0.27	0.1

**Table 5 microorganisms-08-01273-t005:** Results for the ANOSIM applied on the normalized and resembled (Euclidean distances) data for each sub-area.

		Prokaryotic Activity		Physical-Chemical Characteristics		Prokariotic Abund. and Biom.	OM		OM Ratios			
		R Statistic	Sign. %	R Statistic	Sign. %	R Statistic	Sign. %	R Statistic	Sign. %		R Statistic	Sign. %
global test		0.24	0.1	0.17	0.1	0.34	0.1	0.13	0.4	0.21	0.1	
TNB-W	TNB-mid	0.23	0.5	0.00	37.3	0.05	14.5	−0.05	90.3	−0.05	83.6	
	TNB-E	0.09	14.9	0.05	23	0.10	10.7	−0.02	56.8	0.03	31.6	
	RS-S	0.40	0.1	0.00	40.5	0.15	0.9	0.20	1.00	0.22	0.3	
	RS-N	0.09	3.4	0.30	0.1	0.54	0.1	0.18	0.50	0.27	0.1	
TNB-mid	TNB-E	0.05	19.7	0.00	33.1	−0.06	69	−0.06	80.9	0.19	3.0	
	RS-S	0.80	0.1	0.15	4.8	0.20	0.7	0.26	0.30	0.33	0.3	
	RS-N	0.19	1.4	0.25	0.4	0.54	0.1	0.22	0.60	0.26	0.2	
TNB-E	RS-S	0.63	0.1	0.26	1.8	0.15	6.6	0.39	0.20	0.26	1.2	
	RS-N	0.05	26.3	0.25	1.9	0.54	0.1	0.20	1.90	0.41	0.1	
RS-S	RS-N	0.19	0.9	0.28	0.1	0.44	0.1	0.06	16	0.16	2.0	

**Table 6 microorganisms-08-01273-t006:** RDA results: conditional effects for the two macro-area data. Sum: increase total sum of eigenvalues after including new variable. Increase: increase in explained variation due to adding an extra explanatory variable.

Macro-Area	Variable	Sum	Increase	
			F Statistic	*p*-Value
coastal TNB	POC/PN ratio	0.19	7.779	0.002
	PN	0.08	4.224	0.004
	S	0.06	3.027	0.028
	proteins	0.06	2.782	0.054
	OX	0.05	2.214	0.072
	PA	0.04	2.288	0.082
	POC	0.03	1.686	0.164
	protein/carbohydrate ratio	0.02	0.964	0.452
	PB	0.02	0.858	0.470
	carbohydrates	0.01	0.418	0.802
	T	0.00	0.125	0.962
offshore RS	POC	0.21	7.873	0.002
	PA	0.12	5.392	0.002
	S	0.08	3.823	0.010
	protein/carbohydrate ratio	0.06	2.928	0.028
	carbohydrates	0.05	2.575	0.044
	proteins	0.04	2.208	0.070
	PN	0.03	1.450	0.198
	OX	0.02	0.910	0.428
	T	0.01	0.796	0.462
	PB	0.01	0.671	0.592
	POC/PN ratio	0.01	0.602	0.614

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
