# Peer review of "Regulation of Microbial Activity Rates by Organic Matter in the Ross Sea during the Austral Summer 2017"

_microorganisms, 2020, doi:10.3390/microorganisms8091273_

Round 1

Reviewer 1 Report

The manuscript „ Regulation of microbial activity rates by organic matter in the Ross Sea during the austral summer 2017” presented for review is very interesting and thus should be published in the Microorganism journal.

The manuscript text requires the following corrections:

In the last paragraph of Introduction there should not be point lists, but instead - a sentence stating the aim and how it will be achieved.

“We aimed at:

1- testing whether the two investigated area showed spatial similarities or differences in terms of prokaryotic-related variables,

2 - finding out the potential regulation of these variables by POM availability and quality, and relate these findings to the environmental constraints”.

Material and Methods

Fig.1 – there should also be a map showing where this area (Ross Sea) is located – perhaps as an inset in a corner?

There is no information on the quality of the obtained results (in Environmental and biochemical parameters).

Results

I would propose describing first the environmental and biochemical parameters, and then microbiological parameters. Thus would be better than the current: section 3.1 Environmental parameters, then 3.2 & 3.3 Microbiological parameters, and finally 3.4 Particulate Organic Matter.

Correlation analysis and multivariate analyses

Figures illustrating the relationships would be welcome (instead of only tables).

Conclusions

There is no worldwide literature context given for this study. The sentence „Our study provides novel visions on the mechanisms underlying the prokaryotic utilization of organic particles in the epipelagic ocean.” – is rather too little. Following the very interesting discussion, there is a need to present the state of the art literature and contrast the new discovery with it - in the global, not just local context.

Supplementary

There is a need to correct the numbers of significant digits in the presented results (coding them according to the calculated measurement errors). The results are currently given with between two and five significant digits. Why so?

E.g. POC – 167.92 – what is the meaning of the 0.92 difference in this result?

POC

PN

POC/PN

proteins

carbohydrates

µg C/l

µg N/l

µg/l

µg/l

167.92

24.97

6.86

120.46

66.33

76.27

12.15

0.54

62.47

24.64

Author Response

The manuscript „ Regulation of microbial activity rates by organic matter in the Ross Sea during the austral summer 2017” presented for review is very interesting and thus should be published in the Microorganism journal.

The manuscript text requires the following corrections: In the last paragraph of Introduction there should not be point lists, but instead - a sentence stating the aim and how it will be achieved.

The aim was changed

 “We aimed at:

1- testing whether the two investigated area showed spatial similarities or differences in terms of prokaryotic-related variables,

2 - finding out the potential regulation of these variables by POM availability and quality, and relate these findings to the environmental constraints”.

Material and Methods

Fig.1 – there should also be a map showing where this area (Ross Sea) is located – perhaps as an inset in a corner?

There is no information on the quality of the obtained results (in Environmental and biochemical parameters).

Specific standard deviations to infer the environmental variability (namely differences between replicates) were provided in the result section.

Results

I would propose describing first the environmental and biochemical parameters, and then microbiological parameters. Thus would be better than the current: section 3.1 Environmental parameters, then 3.2 & 3.3 Microbiological parameters, and finally 3.4 Particulate Organic Matter.

Done

Correlation analysis and multivariate analyses

Figures illustrating the relationships would be welcome (instead of only tables).

In my opinion, different correlations among microbiological parameters and the other variables as physical-chemical, quantitative and qualitative POM, can be resumed better in a table.

Conclusions

There is no worldwide literature context given for this study. The sentence „Our study provides novel visions on the mechanisms underlying the prokaryotic utilization of organic particles in the epipelagic ocean.” – is rather too little. Following the very interesting discussion, there is a need to present the state of the art literature and contrast the new discovery with it - in the global, not just local context.

Conclusions were changed

Supplementary

There is a need to correct the numbers of significant digits in the presented results (coding them according to the calculated measurement errors). The results are currently given with between two and five significant digits. Why so?

E.g. POC – 167.92 – what is the meaning of the 0.92 difference in this result?

We limited the results to two digits.

POC       PN          POC/PN proteins               carbohydrates

µg C/l    µg N/l                  µg/l        µg/l

167.92   24.97     6.86       120.46   66.33

76.27     12.15     0.54       62.47     24.64

Reviewer 2 Report

Review of ‘Regulation of microbial activity rates by organic 3 matter in the Ross Sea during the  austral summer 2017’ by Zaccone et al.

This manuscript reports on a sampling campaign, conducted in 2 areas in the Antarctic Ross Sea. Each area was divided in sub areas, the general aim of this manuscript was to test whether there would be differences between variables that are important to structure prokaryotic communities, and to investigate the importance of environmental variables in these differences. The sampling scheme is very good, the choice of variables to be measures is very relevant, and the topic of the paper is timely. However, there are some unclarities in the use of the statistics, which should be clarified before the paper can be accepted for publication. Statistics are meant to support the interpretation and statements made in the discussion, and therefore the strategy for data analysis should be beyond any doubt. I don’t question the choice of techniques, as I would have considered similar techniques. However, I am not fully convinced that they were applied correctly. They might have been applied correctly, but this is difficult to judge from the Material and Methods, and the way they are presented in the results section. In addition, I found some parts of the discussion difficult to read, mainly because of the use of rather long sentences. I suggest to use shorter sentences, to make sure that the reasoning is easier to follow. I recommend to adapt the manuscript along the lines of my major comments above, and detailed comments below, before it can be accepted for publication.

Line 22-24: needs to be rephrased. Now; it reflects a local finding. Should be a general finding, observed at Terra Nova Bay. Somehting like: the higher/lower quality of the POM lead to an increase in XXX at Terra Nova Bay.

Figure 1: the map should have a 'North' arrow.

Line 193: this seems as a univariate approach, while at the end of the para, the authors describe a multivariate strategy to relate prokaryotic activities to a set of explanatory variables. I have the feeling that the latter technique is the technique that matches the objectives of the manuscript. The authors should carefully consider the necessity of the correlation approach.

Line 197: the authors use ANOSIM to detect differences in multivariate patterns, which is ok. However, it is not clear how the ANOSIM is performed, and how it is interpreted. From what is written in the Material and Methods, I assumed that the authors would test for differences between areas. Looking at the table with the results, it seems as if they tested for differences between depths, within an area (which is something different). In addition, the table seems to provide results of pairwise tests, and the text mentions significance levels for these pairwise tests. However, due to the ANOSIM algorithm, the significance level for pairwise tests is not important. Researchers should use the magnitude of R (the test statistic), to decided on larger/smaller differences. In addition, pairwise test should only be reported when the overall test was significant, and this test result is lacking.

Line 206: this line states that SIMPER is used to identify the variables that are responsible for a difference between areas. However, I read the results as if SIMPER was used to identify those variables that are responsible for differences between 'subareas, within an area'. This should be clear from the description here. Furthermore, it only makes sense to check SIMPER for areas that are indeed significantly different, or show large differences (i.e. as shown by ANOSIM, or large R values in the pairwise tests) to avoid overinterpretation of small differences.

General comment on Fig. 2 and all figures on the vertical profiles: the data points should not be connected. I can think of a lot of other lines, connecting the data points...

Line 433, Table 4: Here again, I have doubts on the interpretation of the ANOSIM. The first line shows a test comparing TNB with RS, which seems ok. However, the lines below report pairwise differences, between subgroups of TNB, and subgroups of RS. This requires an overall test for subgroups within TNB. This overall test needs to be significant, before the pairwise test an be consulted. The significance of these pairwise tests is not important, as it depends on the number of replicates. The authors need to investigate the magnitude of R, which is for the section on TNB-W low to very low. It is also very strange to see 'ns' as result for a test statistic, because the software always returns a value for that statistic (at least, if the tested design is valid).

The test for RS-S and RS-N can be the result of an overall test, as only 2 groups are tested here.

Line 441: Table 5 was not included?

Line 451: please clarify: significant differences in what?

Line 494: to be correct: the role of the physical-chemical variables WITHIN AN AREA was highlighted

Line 567: e =>a

Author Response

This manuscript reports on a sampling campaign, conducted in 2 areas in the Antarctic Ross Sea. Each area was divided in sub areas, the general aim of this manuscript was to test whether there would be differences between variables that are important to structure prokaryotic communities, and to investigate the importance of environmental variables in these differences. The sampling scheme is very good, the choice of variables to be measures is very relevant, and the topic of the paper is timely. However, there are some unclarities in the use of the statistics, which should be clarified before the paper can be accepted for publication. Statistics are meant to support the interpretation and statements made in the discussion, and therefore the strategy for data analysis should be beyond any doubt. I don’t question the choice of techniques, as I would have considered similar techniques. However, I am not fully convinced that they were applied correctly. They might have been applied correctly, but this is difficult to judge from the Material and Methods, and the way they are presented in the results section. In addition, I found some parts of the discussion difficult to read, mainly because of the use of rather long sentences. I suggest to use shorter sentences, to make sure that the reasoning is easier to follow. I recommend to adapt the manuscript along the lines of my major comments above, and detailed comments below, before it can be accepted for publication.

Line 22-24: needs to be rephrased. Now; it reflects a local finding. Should be a general finding, observed at Terra Nova Bay. Somehting like: the higher/lower quality of the POM lead to an increase in XXX at Terra Nova Bay.

Done

Figure 1: the map should have a 'North' arrow.

Figure 1 was improved.

Line 193: this seems as a univariate approach, while at the end of the para, the authors describe a multivariate strategy to relate prokaryotic activities to a set of explanatory variables. I have the feeling that the latter technique is the technique that matches the objectives of the manuscript. The authors should carefully consider the necessity of the correlation approach.

The text was, actually, misleading and we changed it. We highlighted that, first, a univariate approach was used (one-way ANOVA) to find out significant differences between sites for the same variable. Correlation analysis was maintained, because it helped to explain some specific trends related to EEAs.

Line 197: the authors use ANOSIM to detect differences in multivariate patterns, which is ok. However, it is not clear how the ANOSIM is performed, and how it is interpreted. From what is written in the Material and Methods, I assumed that the authors would test for differences between areas. Looking at the table with the results, it seems as if they tested for differences between depths, within an area (which is something different). In addition, the table seems to provide results of pairwise tests, and the text mentions significance levels for these pairwise tests. However, due to the ANOSIM algorithm, the significance level for pairwise tests is not important. Researchers should use the magnitude of R (the test statistic), to decided on larger/smaller differences. In addition, pairwise test should only be reported when the overall test was significant, and this test result is lacking.

All the observations provided here by the Reviewer were considered for the following Table 4, that was similarly commented (see below). Actually, Material & Method section did not report the exact method for the application of ANOSIM in this specific case (vertical stratification). We removed the ANOSIM for the vertical differentiation of the water column in the two macro-areas. In the text, instead, the trends of the main physical characteristics (S and T) were briefly commented to highlight that stratification was shallower and more evident in TNB, deeper and slighter in RS, in agreement with the trends of the other variables.

Line 206: this line states that SIMPER is used to identify the variables that are responsible for a difference between areas. However, I read the results as if SIMPER was used to identify those variables that are responsible for differences between 'subareas, within an area'. This should be clear from the description here. Furthermore, it only makes sense to check SIMPER for areas that are indeed significantly different, or show large differences (i.e. as shown by ANOSIM, or large R values in the pairwise tests) to avoid over interpretation of small differences.

The text was corrected, we used SIMPER to highlight which of the variables were mainly responsible of the similarity within TNB macro-area and within RS macro-area.  

General comment on Fig. 2 and all figures on the vertical profiles: the data points should not be connected. I can think of a lot of other lines, connecting the data points...

We preferred to maintain the lines. Many depths were measured for physical -chemical parameters, we only reported four points for similarity with other figures.

Line 433, Table 4: Here again, I have doubts on the interpretation of the ANOSIM. The first line shows a test comparing TNB with RS, which seems ok. However, the lines below report pairwise differences, between subgroups of TNB, and subgroups of RS. This requires an overall test for subgroups within TNB. This overall test needs to be significant, before the pairwise test can be consulted. The significance of these pairwise tests is not important, as it depends on the number of replicates. The authors need to investigate the magnitude of R, which is for the section on TNB-W low to very low. It is also very strange to see 'ns' as result for a test statistic, because the software always returns a value for that statistic (at least, if the tested design is valid). The test for RS-S and RS-N can be the result of an overall test, as only 2 groups are tested here.

The Material & Methods section was corrected. We performed first an analysis comparing TNB and RS for each of the five matrices of data. Then, we performed another ANOSIM, considering, within each matrix, the sub-areas, to highlight whether further differences could be find inside each macro-area. To limit the number of tables we placed together the results of the two tests, but it created confusion. Therefore, the two results were given separately and better explained.

All the results were reported, also those that were erroneously considered not significant for the low value of R.

Line 441: Table 5 was not included?

Mistake, the table was now included.

Line 451: please clarify: significant differences in what?

The sentence was changed and the subject specified.

Line 494: to be correct: the role of the physical-chemical variables WITHIN AN AREA was highlighted

Done

Line 567: e =>a

Done